# PAR4-Mediated PI3K/Akt and RhoA/ROCK Signaling Pathways Are Essential for Thrombin-Induced Morphological Changes in MEG-01 Cells

**DOI:** 10.3390/ijms23020776

**Published:** 2022-01-11

**Authors:** Yunkyung Heo, Hyejin Jeon, Wan Namkung

**Affiliations:** Yonsei Institute of Pharmaceutical Sciences, College of Pharmacy, Yonsei University, 85 Songdogwahak-ro, Yeonsu-gu, Incheon 21983, Korea; ykheo107@naver.com (Y.H.); isy0803@naver.com (H.J.)

**Keywords:** MEG-01, platelet, thrombin, PAR1, PAR4, morphological change

## Abstract

Thrombin stimulates platelets via a dual receptor system of protease-activated receptors (PARs): PAR1 and PAR4. PAR1 activation induces a rapid and transient signal associated with the initiation of platelet aggregation, whereas PAR4 activation results in a prolonged signal, required for later phases, that regulates the stable formation of thrombus. In this study, we observed differential signaling pathways for thrombin-induced PAR1 and PAR4 activation in a human megakaryoblastic leukemia cell line, MEG-01. Interestingly, thrombin induced both calcium signaling and morphological changes in MEG-01 cells via the activation of PAR1 and PAR4, and these intracellular events were very similar to those observed in platelets shown in previous studies. We developed a novel image-based assay to quantitatively measure the morphological changes in living cells, and observed the underlying mechanism for PAR1- and PAR4-mediated morphological changes in MEG-01 cells. Selective inhibition of PAR1 and PAR4 by vorapaxar and BMS-986120, respectively, showed that thrombin-induced morphological changes were primarily mediated by PAR4 activation. Treatment of a set of kinase inhibitors and 2-aminoethoxydiphenyl borate (2-APB) revealed that thrombin-mediated morphological changes were primarily regulated by calcium-independent pathways and PAR4 activation-induced PI3K/Akt and RhoA/ROCK signaling pathways in MEG-01 cells. These results indicate the importance of PAR4-mediated signaling pathways in thrombin-induced morphological changes in MEG-01 cells and provide a useful in vitro cellular model for platelet research.

## 1. Introduction

Platelet activation is an essential event in hemostasis and thrombosis [1]. While platelets circulate freely under resting conditions, once vascular injury occurs, platelets are activated by locally generated agonists, including thrombin, ADP, and thromboxane A_2_ (TxA_2_) [2]. Among them, thrombin is the most potent agonist for platelet activation [1]. It stimulates platelets through a subset of G protein-coupled receptors (GPCRs) known as protease-activated receptors (PARs) [3]. Human platelets express two subtypes of PARs, PAR1 and PAR4, that act as thrombin receptors [3]. These two receptors are activated upon cleavage of their N-terminus by thrombin and initiate the multiple G protein-mediated intracellular events that are ultimately required for platelet adhesion and aggregation [4]. PAR-mediated signaling transductions mainly include G_q_-mediated phospholipase C (PLC) activation that triggers intracellular calcium increase and G_12/13_-mediated Rho activation that triggers platelet shape change, and these two synergistically contribute to platelet granule secretion and ultimately result in platelet adhesion and aggregation [5,6].

Studies of the past decades have shown that PAR1 and PAR4 are activated by thrombin, but their activation and signaling mechanisms are different [7]. PAR1 is a transmembrane protein with 425 amino acids including a thrombin cleavage site between Arg41 and Ser42 (TLDPR^41^/^42^SFLLRN) and a hirudin-like thrombin binding domain between residue 53 and 64, and the high-affinity thrombin binding domain promotes the efficient cleavage of PAR1 by thrombin [4,8]. In contrast, PAR4 is cleaved by thrombin at a specific site between Arg47 and Gly48 (LPAPR^47^/^48^GYPGQV), but because it lacks a hirudin-like domain, it requires ~10-fold higher concentrations of thrombin for activation [7,9,10]. Moreover, PAR1 and PAR4 signal with distinct kinetics, probably due to differences in their shut off mechanism by receptor phosphorylation and internalization [11]. For example, PAR1 activation induces a rapid and transient intracellular calcium signal, whereas PAR4 mediates a slow and sustained intracellular calcium signal [11,12,13]. In addition, PAR4 induces a more prolonged Rho-dependent signal, resulting in greater procoagulant phenotypes such as factor V (FV) secretion and microparticle generation, compared with PAR1 activation [14]. Differences in the receptor affinity, signaling kinetics, and procoagulant phenotypes of PAR1 and PAR4 for thrombin are related to the fact that PAR1 primarily initiates hemostasis at low levels of locally generated thrombin, whereas PAR4 is required for stable thrombus formation at strongly increased local thrombin concentrations [14,15,16].

Meanwhile, platelet shape change is the initial response to G_12/13_-mediated RhoA activation and is the hallmark of platelet activation [17,18]. Under resting conditions, platelets remain discoid-shaped, and once activated, they change to a more spherical form with pseudopodial protrusions mediated by cytoskeleton reorganization [19]. The G_12/13_-dependent RhoA pathway that mediates platelet shape change is also involved in platelet adhesion and secretion, and the deletion of either G_α13_ or Rho GTPases results in severe dysfunction in platelets [6,20,21]. These studies imply that thrombin-induced shape change significantly contributes to platelet function. Although the physiological roles of PAR1 and PAR4 in thrombin-induced calcium signaling, granule secretion, GPIIb/IIIa activation, and platelet aggregation have been extensively studied, their distinct roles in thrombin-induced shape change have not been clearly elucidated [9,11,12,14,22]. However, platelets that are easily aggregate with agonists are stable for only 5 days, and the functions are variable depending on the donor, so an efficient in vitro cellular model to replace platelets is needed [23,24,25]. Therefore, in this study, we established an in vitro cellular model for thrombin-induced morphological changes in platelets using MEG-01 cells, a human megakaryoblastic leukemia cell line.

MEG-01 cells are derived from the bone marrow of a patient with blast crisis of chronic myelogenous leukemia [26]. MEG-01 cells exhibit phenotypic properties that closely resemble those of megakaryoblasts; platelet GPIIb/IIIa antigen was demonstrated, while none of the markers for B cells (CD9, CD10, CD19) or T cells (CD2, CD5, CD6, CD7) were detected [26]. Moreover, MEG-01 cells have been shown to differentiate to mature megakaryocytes (MKs) in the presence of phorbol diesters and release platelet-like particles, and these differentiated forms showed a profound enhancement in expression of several MK-platelet-specific proteins including GPIIb/IIIa, fibrinogen, von Willebrand factor, factor XIIIa, β-thromboglobulin, and HLA class I antigen [27,28]. Based on these well-established megakaryoblastic phenotypes, MEG-01 cells have been commonly used for studies on differentiation of MKs and platelet-like particle production [29,30,31,32,33]. Recently, MEG-01 cells were established as a useful in vitro cellular model for platelet calcium signaling [34].

In this study, we observed the effect of PAR1 and PAR4 activation on thrombin-induced morphological changes in MEG-01 cells. To quantitatively measure the morphological changes of MEG-01 cells, we developed an image-based method and extensively studied the underlying molecular mechanisms involved in thrombin-induced PAR1 and PAR4 activation and morphological changes in MEG-01 cells.

## 2. Results

### 2.1. PAR1 and PAR4 Are Endogenously Expressed in MEG-01 Cells

PAR1 and PAR4 play critical roles in thrombin-induced human platelet activation and aggregation and have been recognized as novel therapeutic targets for antithrombotic therapy. In this study, we confirmed the expression of both thrombin receptors in a megakaryoblastic cell line, MEG-01. RT-PCR analysis revealed the mRNA expression of PAR1 and PAR4 in MEG-01 cells (Figure 1A). In order to confirm that there was no contamination of genomic DNA in the RNA extract, a group not treated with reverse transcriptase was added. Western blot analysis was performed for the detection of PAR1 protein expression. Cell lysates of human melanoma A2058 cells that exhibit high expression of PAR1, and PAR1 knocked out (K/O) A2058 cells, were used as positive and negative controls to confirm the specificity of the antibody for PAR1. Although the expression level was weak compared to A2058 cells, protein expression of PAR1 was clearly identified in MEG-01 cells (Figure 1B).

### 2.2. Functional Expression of PAR1 and PAR4 in MEG-01 Cells

Previous studies have shown differential calcium signaling by PAR1 and PAR4 in response to thrombin in platelets [10,11,35]. Intracellular calcium signaling profiles of PAR1 and PAR4 in MEG-01 cells were characterized using thrombin, PAR1-activating peptide (AP), PAR4-AP, and selective antagonists of PAR1 (vorapaxar) and PAR4 (BMS-986120). As shown in Figure 2A, PAR1-AP induced a rapid and transient increase in intracellular calcium level, which was completely inhibited by vorapaxar. PAR4-AP, on the other hand, induced slower but prolonged calcium signal that was fully blocked by BMS-986120 (Figure 2B). Interestingly, a low concentration of thrombin (0.5 U/mL)-induced calcium signal was almost completely blocked by a PAR1 inhibitor, namely vorapaxar (Figure 2C). However, a higher concentration of thrombin (5 U/mL)-induced calcium signal was not completely blocked by the inhibition of PAR1 (Figure 2D). Notably, in thrombin-induced calcium signaling, PAR4 inhibition by BMS-986120 showed a calcium signal similar to that of PAR1-AP, and PAR1 inhibition by vorapaxar showed a calcium signal similar to PAR4-AP. Furthermore, the combined inhibition of PAR1 and PAR4 resulted in complete inhibition of the thrombin-induced calcium signal (Figure 2D). These results showed that MEG-01 cells functionally express both PAR1 and PAR4, which show distinct calcium signaling by thrombin.

### 2.3. PAR1- and PAR4-Mediated Morphological Changes in MEG-01 Cells

We further investigated the thrombin-mediated cellular responses in MEG-01 cells. Interestingly, the activation of PAR1 and PAR4 induced morphological changes in MEG-01 cells that were similar to the shape change observed in activated platelets. The shape change of platelets in response to various platelet activators, such as thrombin and ADP, is essential for their spreading and stable adhesion, and requires remodeling of the actin cytoskeleton, which is regulated by complex signaling pathways [36]. To confirm whether the morphological changes observed in MEG-01 cells involve actin cytoskeleton remodeling, activated cells were labeled with phalloidin for F-actin staining. In resting condition, the MEG-01 cells had rounded shapes with few pseudopods, but when activated by thrombin, PAR1-AP, or PAR4-AP, the cells showed dynamic morphological changes to a spherical shape with multiple pseudopods (Figure 3A).

The platelet-like morphological changes were quantitatively analyzed using circularity measurement. For the analysis, 20 cells were randomly selected for each group. The mean value of the cell circularity in resting condition was 0.86 ± 0.08. When applied with thrombin (0.1 U/mL), PAR1-AP, and PAR4-AP, the mean value was significantly decreased to 0.67 ± 0.14, 0.70 ± 0.17, and 0.59 ± 0.16, respectively. Note that PAR4-AP, which triggered a lower maximum calcium peak, induced morphological changes more strongly compared to PAR1-AP (Figure 3B). The cell diameters were not significantly affected by PAR1 and PAR4 activation (Figure 3C).

### 2.4. Assessment of PAR1- and PAR4-Mediated Morphological Changes in Living MEG-01 Cells

We developed a novel and convenient live cell image-based assay to extensively study and compare the morphological changes induced by PAR1 and PAR4 activation in MEG-01 cells. As shown in Figure 4A, non-adherent MEG-01 cells were gently suspended in HEPES buffered saline and were allowed to adhere to glass bottom microplates for 30 min. Cells were then treated with calcein-AM-containing solution with or without inhibitors for 30 min, and the stained cells were imaged 30 min after treatment of PAR1-AP, PAR4-AP, or thrombin. Representative bright field and calcein-AM-stained cell images are shown in Figure 4B. Circularity measurements obtained using calcein-AM were not as accurate as those obtained using phalloidin because calcein-AM stained the cytoplasm, but calcein-AM was suitable for staining living cells and was sufficient to measure circularity changes caused by PAR1 and PAR4 activation.

PAR1 and PAR4 activation significantly reduced the circularity of MEG-01 cells, and the reduction in circularity by PAR1 and PAR4 was completely blocked by pretreatment with vorapaxar and BMS-986120, respectively (Figure 4C,D). In addition, thrombin-induced morphological change was completely blocked when vorapaxar and BMS-986120 were simultaneously applied. Notably, while vorapaxar inhibited only 50% of thrombin-induced shape change, BMS-986120 suppressed 80% of it (Figure 4E). These results indicated that PAR4 contributes more significantly to thrombin-induced morphological changes than PAR1 in MEG-01 cells.

### 2.5. Prolonged Effect of PAR4-Mediated Morphological Changes in MEG-01 Cells

A time-lapse experiment was performed to investigate the time-dependent effects of PAR1 and PAR4 activation on morphological changes in MEG-01 cells. Remarkably, the morphological changes induced by PAR1-AP were partially reversible, whereas the morphological changes induced by PAR4-AP and thrombin were prolonged for the entire time observed (Figure 5). In addition, PAR4-AP induced a stronger and prolonged decrease in circularity compared to PAR1-AP in MEG-01 cells (Figure 5C,D). These results correlate with the differences in calcium signaling by PAR1 and PAR4 in that PAR4 activation induced longer-duration calcium signaling, while PAR1 activation induced transient calcium signaling in MEG-01 cells.

### 2.6. The Morphological Changes of MEG-01 Cells by PAR4 Activation Are Calcium-Independent and Strong, but PAR1 Activation Is Not

Actin cytoskeleton remodeling is known to be either calcium-dependent or calcium-independent [18,37]. To further investigate whether the morphological change induced by thrombin in MEG-01 cells is a calcium-dependent or independent event, MEG-01 cells were treated with 2-APB, an antagonist of the IP_3_ receptor, and ionomycin, a calcium ionophore. 2-APB inhibited PAR1-AP- and PAR4-AP-induced intracellular calcium increases in a dose-dependent manner, and completely inhibited the intracellular calcium increase at 100 μM (Figure 6A,B). Remarkably, the complete inhibition of calcium release by 2-APB significantly suppressed PAR1-AP-induced morphological changes in MEG-01 cells (Figure 6D). Interestingly, however, complete inhibition of calcium signaling by 100 μM of 2-APB enhanced PAR4-AP-induced morphological changes (Figure 6E). These results revealed that PAR1 and PAR4 regulate morphological changes through different signaling pathways. Meanwhile, ionomycin significantly triggered a sustained increase in the intracellular calcium level (Figure 6C), but did not induce morphological changes (Figure 6F). These results indicated that morphological changes in MEG-01 cells are predominantly regulated through the downstream signaling pathways of PAR1 and PAR4, rather than through the calcium signaling itself.

### 2.7. Thrombin-Induced Morphological Change Is Primarily Mediated through PI3K-Akt and Rho-ROCK Pathway via PAR4 Activation

To elucidate the underlying molecular mechanisms of PAR1- and PAR4-mediated morphological changes in MEG-01 cells, we performed high-content screening (HCS) with a chemical library containing 200 kinase inhibitors. HCS was conducted separately for PAR1 and PAR4 using PAR1-AP and PAR4-AP, respectively. As shown in Figure 7A,B, several inhibitors targeting various kinases including receptor tyrosine kinases (RTKs), PI3K, Akt, mTOR, and Rho-associated protein kinase (ROCK) were identified to inhibit the morphological changes induced by PAR1 activation. PAR4 activation-induced morphological changes in MEG-01 cells were effectively blocked by several inhibitors of PI3K, Akt, and ROCK. The representative inhibitor for each target kinase was selected and used for further investigation. Among the eight RTK inhibitors, tivozanib, a selective VEGFR inhibitor, was selected because VEGFR2 was the most repeatedly targeted. For inhibitors of PI3K, Akt, mTOR, and ROCK, the compounds with the highest potency, XL147, AT7867, PP242, and netarsudil, were selected, respectively. PAR1-AP-induced morphological changes were significantly blocked by tivozanib, XL147, AT7867, PP242, and netarsudil, but PAR4-AP-induced morphological changes were significantly blocked by only XL147, AT7867, and netarsudil (Figure 7C–E). Remarkably, thrombin-induced morphological changes in MEG-01 cells were strongly blocked by XL147, AT 7867, and netarsudil, which was consistent with PAR4-AP. (Figure 7E,F). These results indicate that PAR4-mediated cellular signaling pathways predominate over PAR1 signaling pathways in thrombin-induced morphological changes in MEG-01 cells, as summarized in Figure 8.

## 3. Discussion

Platelet activation is crucial for hemostasis and thrombosis, and signaling within platelets begins via the activation of PARs, P2Y12, and the thromboxane receptor by thrombin, ADP, and TxA_2_, respectively [1]. Signaling events underlying the platelet activation induced by these agonists commonly include the G_q_-mediated activation of PLC, which triggers intracellular calcium increase and granule secretion, G_12/13_-mediated RhoA activation, which stimulates the reorganization of the actin cytoskeleton, resulting in the formation of filopodia and lamellipodia, and PI3K activation, which is crucial for stabilizing platelet aggregates [1,5]. Among these agonists, thrombin most potently initiates human platelet activation by stimulating PAR1 and PAR4 [1].

In this study, we found a close similarity between human platelets and the human megakaryoblastic leukemia cell line (MEG-01) in terms of thrombin-induced (1) calcium signaling and (2) morphological changes. Previous studies have shown that PAR1- and PAR4-mediated calcium signaling display distinct kinetics and magnitudes in human platelets in response to thrombin. While PAR1 activation induces a rapid and transient increase in intracellular calcium, PAR4 activation results in a slower but more sustained signal in platelets [11]. As shown in Figure 2, in MEG-01 cells, PAR1 activation induced a calcium spike that shortly returned to the baseline, while PAR4 activation triggered a sustained calcium increase, which did not completely return to the baseline during the entire time measured. Moreover, PAR4-mediated calcium signaling in MEG-01 was achieved only at high concentrations of thrombin, as demonstrated in human platelets in a previous study [38].

Platelets rapidly undergo shape change from discoid to spherical upon activation. At resting condition, platelets maintain their discoid shape and dynamically rearrange their cytoskeleton in response to thrombin, and the platelet shape change is considered to be a prerequisite for platelet aggregation [19,36]. According to previous studies, platelet shape change is regulated by both calcium-dependent and independent signaling pathways [18]. G_q_-mediated Ca^2+^/calmodulin-dependent pathways and G_12/13_-mediated RhoA/ROCK signaling pathways independently promote myosin light chain phosphorylation and induce dynamic cytoskeleton remodeling in platelets [18,39]. In this study, we investigated the underlying mechanism by assessing the effects of the IP_3_ receptor antagonist 2-APB and a set of kinase inhibitors on PAR1- and PAR4-mediated morphological changes and found that ROCK inhibitors commonly suppressed morphological changes upon PAR1 and PAR4 activation (Figure 7). On the other hand, the inhibition of cytosolic calcium release by 2-APB only suppressed PAR1-AP-induced morphological changes (Figure 6D). Interestingly, the partial inhibition of the IP_3_ receptor by 10 µM of 2-APB showed no effect on PAR4-mediated morphological changes, whereas the complete inhibition of the IP_3_ receptor by 100 µM of 2-APB enhanced the morphological changes in MEG-01 cells (Figure 6D,E). Although the underlying mechanism of this phenomenon was not elucidated in this study, it may be due to interference in the calcium-induced activation of protein kinase A (PKA) that negatively regulates thrombin-induced shape change in platelets [40].

In the present study, we also provide important information concerning the role of the PI3K-Akt pathway in thrombin-induced morphological changes in MEG-01 cells. Although the question regarding whether PAR1 and PAR4 directly activate PI3K is still unresolved, they apparently activate the PI3K-Akt pathway [1,41]. Indeed, based on its critical role in modulating platelet aggregation, PI3K has been acknowledged as a potential therapeutic target in thrombosis [42,43]. In Figure 7, we show that both the PI3K inhibitor (XL147) and the Akt inhibitor (AT7867) significantly blocked PAR1- and PAR4-mediated morphological changes in MEG-01 cells, and this result is consistent with previous studies showing that the PI3K-Akt pathway is important for both PAR1- and PAR4-mediated conformational changes in human platelets. In addition, we found an important difference between the PAR1- and PAR4-mediated signaling pathways. Inhibition of RTKs and mTOR completely inhibited the morphological changes induced by PAR1 activation, while having no effect on those induced by PAR4 activation. These results indicate that the PI3K-Akt pathway involved in PAR1-mediated morphological changes has crosstalk between PAR1 and RTK and may also pass through the PI3K/Akt/mTOR pathway. Although it is still uncertain how PAR4 activates the PI3K-Akt pathway, none of the tyrosine kinases that we tested inhibited PAR4-mediated morphological changes. This again suggests that PAR1 and PAR4 mediate thrombin-induced morphological changes through a different set of signaling pathways.

MEG-01 is a megakaryoblastic cell line that is capable of differentiating into MKs and generating platelet-like particles, and it is widely used in megakaryocytopoiesis and in vitro platelet production studies [27,28,30]. During differentiation into MKs, MEG-01 cells show dynamic morphological changes with remodeling of cytoskeletal proteins such as actin, α-tubulin, and β1-tubulin, which are prerequisites for proplatelet formation [33]. Notably, thrombin treatment induces dramatic morphological changes in human MKs that are reminiscent of those found in platelets, such as shape changes and organelle centralization [44]. In addition, it is well known that cytoskeletal remodeling of MKs and platelets is primarily regulated through the RhoA/ROCK signaling pathway [21,45]. Interestingly, these results are consistent with our findings that PAR1-AP and PAR4-AP significantly induce morphological changes in MEG-01 cells (Figure 5), and thrombin-induced morphological changes in MEG-01 cells are predominantly regulated by PAR4-mediated PI3K/Akt and RhoA/ROCK signaling pathways (Figure 7). Therefore, these results strongly suggest that PAR1- and PAR4-mediated signaling can play different roles, not only in terms of morphological changes but also in other related physiological events induced by thrombin in MKs.

Altogether, this study suggests that the PAR4-mediated signaling pathway is more important than the PAR1 signaling pathway in terms of thrombin-induced morphological changes in MEG-01 cells, and MEG-01 cells can be a useful tool in studying PAR4-mediated shape changes in platelets. PAR4 is of great interest as a new potential therapeutic target for antiplatelet drugs because the therapeutic blockade of PAR4 is expected to have a high antithrombotic effect with low bleeding risk [15,16,35,46]. Moreover, since abnormal morphological changes in platelets are associated with pathophysiological conditions, the regulation of platelet cytoskeletal dynamics must be explored in the process of drug discovery for novel PAR4 antagonists [47]. However, platelets used for research purposes are currently donor-derived, which is associated with some important limitations in terms of short shelf life, donor dependency, and donor variability [24,48]. Here, we showed that MEG-01 cells functionally express PAR1 and PAR4, and that thrombin induce morphological changes in MEG-01 cells through PAR1 and PAR4 activation, such as that observed in human platelets. In addition, our circularity measurement method is robust and applicable to living cells. Therefore, MEG-01 cells, in combination with the circularity measurement method, can be a useful tool for studying PAR-mediated signaling pathways in platelets, especially morphological changes occurring through thrombin receptors.

## 4. Materials and Methods

### 4.1. Cell Culture and Cell Lines

All cells were cultured at 37 °C and 5% CO_2_. MEG-01 cells were grown in RPMI1640 medium and A2058 cells were grown in DMEM medium. All media contained 10% FBS, 100 units/mL penicillin and 100 µg/mL streptomycin.

### 4.2. Materials and Reagents

PAR1-AP (TFLLR-NH2) and PAR4-AP (AYPGKF-NH2) were synthesized from Cosmogenetech (Seoul, Korea). Vorapaxar was purchased from Axon Medchem (Groningen, Netherlands), AZ3451 from Tocris Bio-techne (Minneapolis, MN, USA), BMS-986120 from Cayman Chemical (Ann Arbor, MI, USA), and other chemicals, unless otherwise indicated, were purchased from Sigma-Aldrich (St. Louis, MO, USA). The kinase inhibitor library used in HCS was purchased from Targetmol (Wellesley Hills, MA, USA).

### 4.3. Immunoblotting

A2058 and MEG-01 cells were plated and grown overnight on 6-well plates. For sample preparation, cells were lysed with RIPA lysis buffer (EMD Millipopre Corp., Billerica, MA, USA) supplemented with a protease inhibitor cocktail. Whole-cell lysates were centrifuged at 13,000× *g* for 20 min at 4 °C and the concentrations of supernatant protein samples were determined with a Bradford protein kit assay (Thermo Scientific, Waltham, MA, USA). Equal amounts of protein (70 μg protein/lane) were loaded and separated by 4–12% Tris-glycine precast gel (Koma Biotech, Seoul, Korea). The gel was then transferred to a PVDF membrane (Millipore, Billerica, MA, USA). The membranes were washed with TBST (Tris-buffered saline with 0.1% tween 20) and incubated in blocking buffer (5% skim milk in TBST) at room temperature for 1 h. The membranes were incubated overnight at 4 °C with primary antibodies specific for PAR1 and β-actin (Santa Cruz Biotechnology, Inc., Santa Cruz, CA, USA; 1:1000). After three washes with TBST, the membranes were incubated for 1h at room temperature with an anti-mouse secondary antibody (Santa Cruz Biotechnology, Inc., Santa Cruz, CA, USA; 1:5000). The membranes were washed 3 times with TBST for 5 min each and then developed using the ECL Plus immunoblotting detection system (GE Healthcare, Piscataway, NJ, USA).

### 4.4. Reverse Transcription Polymerase Chain Reaction

Total mRNA was isolated using TRIzol™ reagent (Invitrogen, Carlsbad, CA, USA) and the cDNA library was generated by reverse transcription using the PrimeScript™ 1st strand cDNA synthesis kit (Takara, Tokyo, Japan) following the manufacturer’s instructions. The reverse transcription reaction conditions were 30 °C for 10 min, 42 °C for 60 min, and 95 °C for 5 min, with or without reverse transcriptase. The RT products (1 μL) were amplified by PCR. Reaction conditions consisted of 95 °C for 5 min, 40 cycles of 95 °C for 45 s, 55 °C for 20 s, 72 °C for 60 s, and 72 °C for 3 min. The primers used for PCR were as follows. PAR1, Sense-strand primer: CAG TTT GGG TCT GAA TTG TGT CG. Antisense-strand primer: TGC ACG AGC TTA TGC TGC TGA C. Resulting PCR product: 592 bp. PAR4, Sense-strand primer: AAC CTC TAT GGT GCC TAC GTG C. Antisense-strand primer: CCA AGC CCA GCT AAT TTT TG. Resulting PCR product: 542 bp. ACTB, Sense-strand primer: GAGAAGATGACCCAGATCATG. Antisense-strand primer: ATCACGATGCCAGTGGTAC. Resulting PCR product: 135 bp. The PCR products were loaded in 2.5% agarose gel.

### 4.5. Intracellular Calcium Measurement

MEG-01 cells were plated on a 96-well clear bottom black wall plate (Corning Inc., Corning, NY, USA). The cells were loaded with Fluo-4 NW (Invitrogen, Carlsbad, CA, USA) according to the manufacturer’s instructions. After 1 h incubation, the cells were treated with inhibitors for 10 min and fluorescence was measured using the FLUOstar Omega microplate reader (BMG LABTECH, Offenburg, Germany). PAR1- and PAR4-activating peptide (AP), as well as thrombin, were injected to the cells by the microplate reader at 5 s after the start of fluorescence reading.

### 4.6. Immunofluorescence Staining

Non-adherent MEG-01 cells were gently suspended in HEPES buffered saline and were allowed to adhere to 8-chamber microscope slides for 30 min. Seeded cells were treated with corresponding agonists for 30 min and the cells were fixed with 4% formaldehyde for 10 min at RT, washed with PBS, and permeabilized with 0.5% Triton X-100 for 10 min. After three washes with PBS, the slides were incubated with blocking buffer (1% BSA, PBS) for 30 min and then with FITC-phalloidin for 1h at RT. After another three washes with PBS, the slides were stained with DAPI for 10 min, and were washed. Finally, the slides were mounted and observed using the LionHeart FX automated microscope with a 40× objective and with the excitation/emission wavelengths [49] set to 469/525 and 377/447 for GFP and DAPI, respectively.

### 4.7. Calcein-AM-Based Assessment of Morphological Change

Non-adherent MEG-01 cells were gently suspended in HEPES buffered saline and were allowed to adhere to glass bottom microplates for 30 min. Seeded cells were then treated with a solution containing Calcein-AM with or without inhibitors for 30 min. The stained cells were imaged 30 min after treatment of corresponding agonists. Imaging and data analysis (measurement of circularity and cell diameter) were conducted using the Lionheart FX automated microscope (BioTek Instruments, Winnooski, VT, USA).

### 4.8. Data and Statistical Analysis

For all statistically analyzed studies, experiments were performed at least three times independently. The results for multiple trials are presented as the mean ± standard deviation (S.D.). Statistical analysis was performed using the Student’s *t*-test. A value of *p* < 0.05 was considered statistically significant.

## Figures and Tables

**Figure 1 ijms-23-00776-f001:**
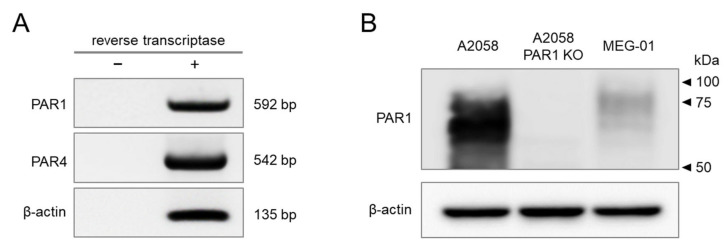
Endogenous expression of PAR1 and PAR4 in MEG-01 cells. (**A**) RT-PCR analysis for PAR1 and PAR4 in MEG-01 cells. PCR products were detected at expected product sizes as indicated. (**B**) Western blot analysis for PAR1 in MEG-01 cells. A2058 and PAR1 knocked out (KO) A2058 cells were used as positive and negative controls, respectively. PAR1 was knocked out by CRISPR/Cas9 in A2058 cells. All experiments were repeated three times.

**Figure 2 ijms-23-00776-f002:**
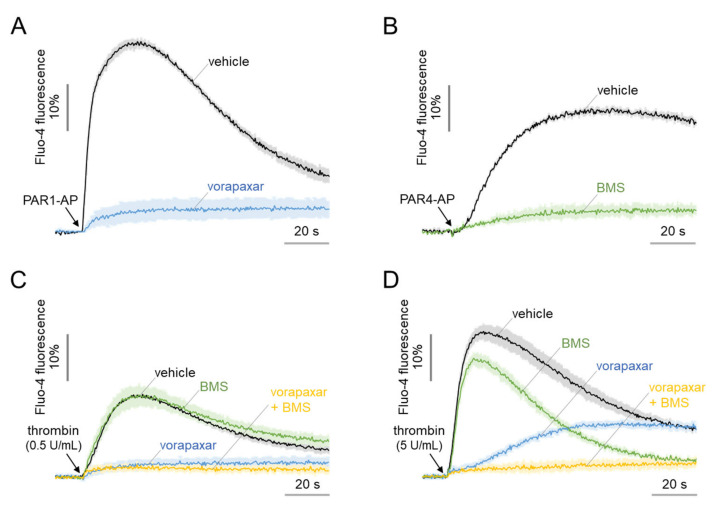
Intracellular calcium increase in response to the activation of PAR1 and PAR4 in MEG-01 cells. Intracellular calcium increase in MEG-01 cells induced by (**A**) PAR1-AP (100 μM), (**B**) PAR4-AP (100 μM), (**C**) low-dose thrombin (0.5 U/mL), and (**D**) high-dose thrombin (5 U/mL). PAR1 and PAR4 were completely blocked by vorapaxar and BMS-986120 (BMS), respectively. Vorapaxar and BMS were applied at 10 μM for 10 min prior to treatment of PAR1-AP, PAR4-AP and thrombin (mean ± S.D., *n* = 3–5).

**Figure 3 ijms-23-00776-f003:**
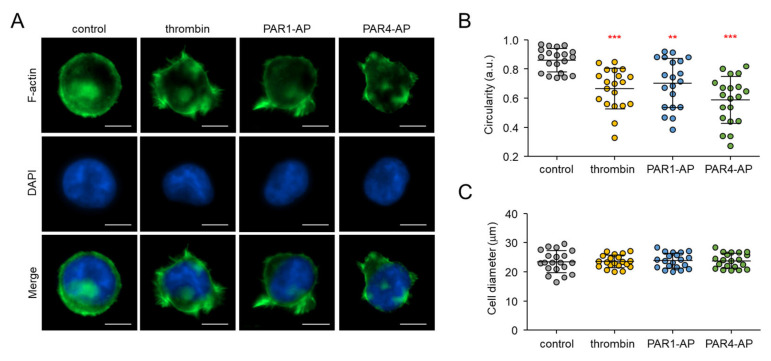
Morphological changes induced by PAR1 and PAR4 activation in MEG-01 cells. (**A**) Representative images of F-actin-stained MEG-01 cells. MEG-01 cells were treated with PAR1-AP (100 μM), PAR4-AP (100 μM), or thrombin (0.1 U/mL) for 30 min and fixed, and then labeled with phalloidin-FITC (green). Nuclei were stained with DAPI (blue). Scale bar = 10 μm. (**B**) Measurement of the circularity of MEG-01 cells. The circularity was measured using Lionheart software. Each dot corresponds to individual analyzed cells. A total of 20 cells were analyzed for each group (mean ± S.D., *n* = 20). (**C**) Measurement of diameter of MEG-01 cells. The diameter was measured using Lionheart software. For each group, 20 cells were analyzed (mean ± S.D., *n* = 20); ** *p* < 0.01, *** *p* < 0.001.

**Figure 4 ijms-23-00776-f004:**
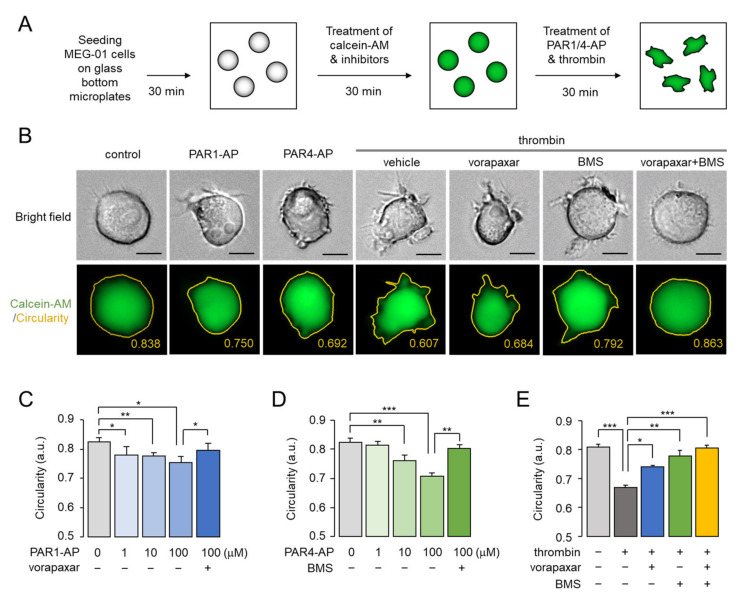
PAR1- and PAR4-mediated morphological changes in living MEG-01 cells. (**A**) Schematic diagram of experimental setup. (**B**) Representative images of MEG-01 cells used for analysis of circularity measurements. Cells were stained with calcein-AM (1 μg/mL) and pretreated with 10 μM of vorapaxar and BMS-986120 (BMS), and then PAR1-AP (100 μM), PAR4-AP (100 μM), or thrombin (0.1 U/mL) were applied for 30 min. Scale bar = 10 μm. (**C**–**E**) Summary of circularity. PAR1 and PAR4 were inhibited with vorapaxar and BMS-986120 (BMS) at the indicated concentrations, respectively. Thrombin-induced activation of PAR1 and PAR4 was blocked by vorapaxar (10 μM), BMS-986120 (10 μM), or both, respectively. Images were captured automatically and circularity was measured using Lionheart software (mean ± S.D., *n* = 4–5). * *p* < 0.05, ** *p* < 0.01, *** *p* < 0.001.

**Figure 5 ijms-23-00776-f005:**
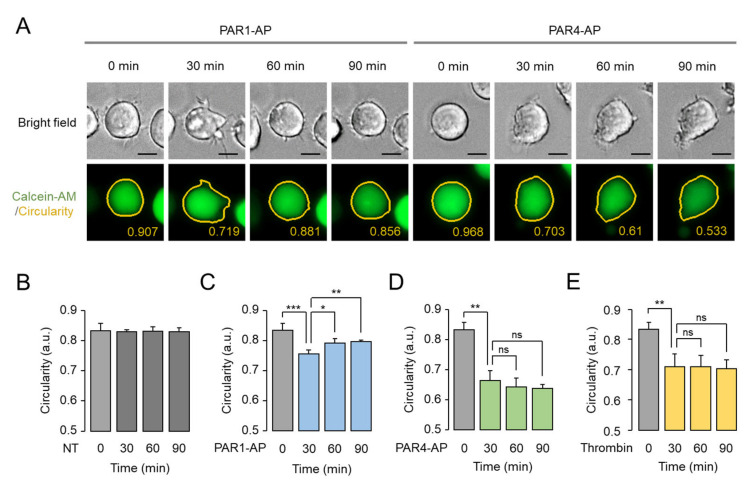
Time-course assessment of PAR1- and PAR4-mediated morphological changes. (**A**) Representative time-lapse images of MEG-01 cells. Cells were stained with calcein-AM (1 μg/mL) and treated with PAR1-AP (100 μM) and PAR4-AP (100 μM). Images were automatically captured by Lionheart every 30 min. Scale bar = 10 μm. (**B**–**E**) Summary of circularity. Cells were applied with PAR1-AP, PAR4-AP, and 0.1 U/mL of thrombin, and circularity was measured in time-course using Lionheart software (mean ± S.D., *n* = 4). * *p* < 0.05, ** *p* < 0.01, *** *p* < 0.001, ns: not significant. Not treated (NT).

**Figure 6 ijms-23-00776-f006:**
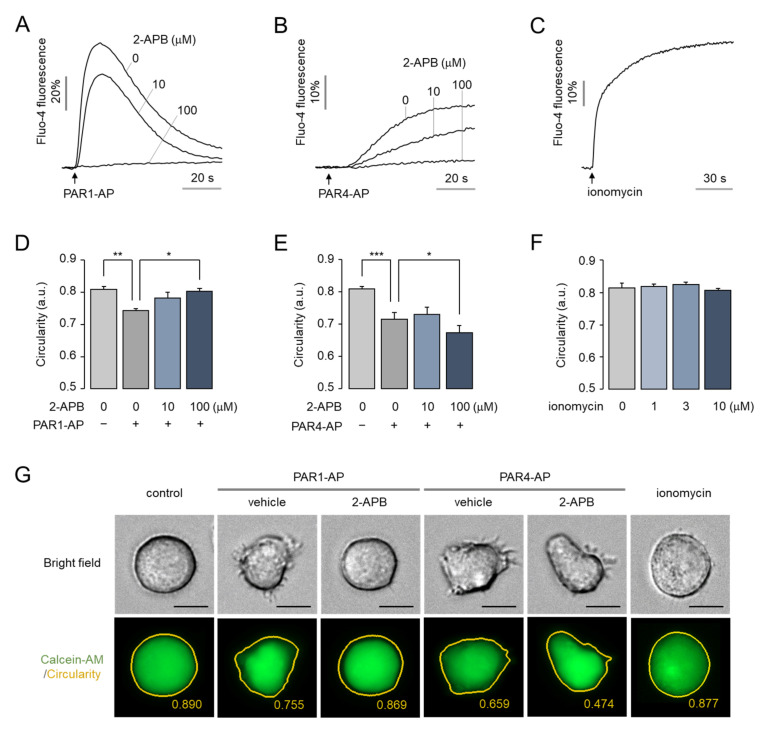
Effect of intracellular calcium signaling on morphological changes in MEG-01 cells. (**A**,**B**) Inhibition of PAP1-AP- and PAR4-AP-induced intracellular calcium increase by 2-APB. MEG-01 cells were pre-treated with 2-APB at the indicated concentrations prior to applying 100 μM of PAR1-AP (**A**) or PAR4-AP (**B**). (**C**) Measurement of ionomycin (10 μM)-induced intracellular calcium increase. (**D**–**F**) Summary of morphological changes. MEG-01 cells were pre-treated with 2-APB at the indicated concentrations for 30 min prior to applying 100 μM of PAR1-AP (**D**) or PAR4-AP (**E**), and ionomycin was applied at the indicated concentrations for 30 min (**F**). (**G**) Representative images of MEG-01 cells. Cells were pretreated with 100 μM of 2-APB for 30 min, and then PAR1-AP (100 μM) and PAR4-AP (100 μM) were applied for 30 min or treated with ionomycin (10 μM) for 30 min. Calcein-AM-stained images were captured automatically and circularity was measured using Lionheart software (mean ± S.D., *n* = 4). Scale bar = 10 μm. * *p* < 0.05, ** *p* < 0.01, *** *p* < 0.001.

**Figure 7 ijms-23-00776-f007:**
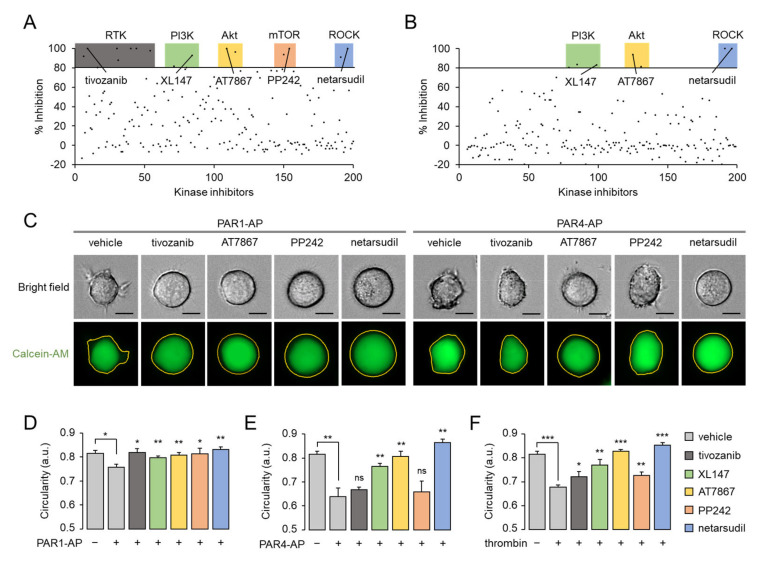
Identification of kinases involved in PAR1- and PAR4-mediated morphological changes in MEG-01 cells. (**A**,**B**) Dot plots of primary screening results for kinase inhibitors that inhibit PAR1- and PAR4-mediated morphological changes. Validated kinase inhibitors that exhibit >80% inhibitory effects on morphological changes are classified according to the class of kinase, and one representative compound for each class is indicated. (**C**) Representative images of MEG-01 cells were stained with calcein-AM (1 μg/mL). Cells were pretreated with 10 μM of indicated kinase inhibitors for 30 min before application of PAR1-AP (100 μM) or PAR4-AP (100 μM) for 30 min. Scale bar = 10 μm. (**D**–**F**) Summary of circularity. Cells were pretreated with 10 μM of indicated kinase inhibitors for 30 min and then PAR1-AP (100 μM), PAR4-AP (100 μM), or thrombin (0.1 U/mL) were applied. Images were taken by Lionheart automatically and circularity was measured using Lionheart software (mean ± S.D., *n* = 4). * *p* < 0.05, ** *p* < 0.01, *** *p* < 0.001.

**Figure 8 ijms-23-00776-f008:**
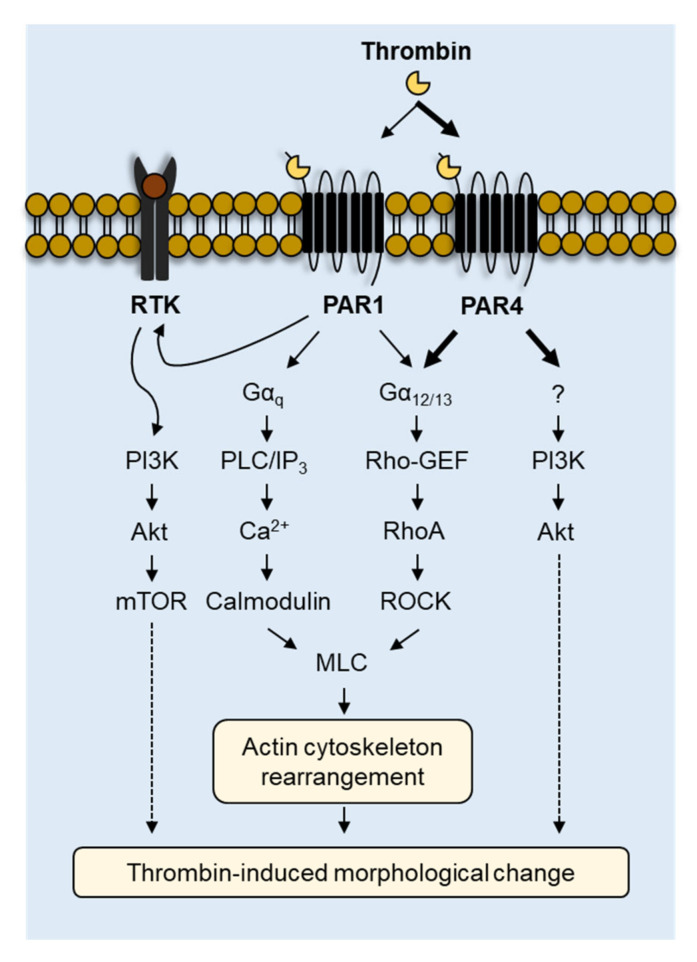
Schematic model of the underlying mechanisms involved in thrombin-induced morphological changes via PAR1 and PAR4 in MEG-01 cells.

## Data Availability

The data that support the findings of this study are available from the corresponding author upon reasonable request.

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
