# Peer review of "PAR4-Mediated PI3K/Akt and RhoA/ROCK Signaling Pathways Are Essential for Thrombin-Induced Morphological Changes in MEG-01 Cells"

_ijms, 2022, doi:10.3390/ijms23020776_

Round 1

Reviewer 1 Report

The paper entitled “PAR4-mediated PI3K/Akt and RhoA/ROCK signaling pathways are essential
for thrombin-induced morphological changes in MEG-01 cells” describes the role of the PAR-4 receptor in the platelet activation mechanism in a new in vitro model. The work is well-structured and properly conducted, even if in literature there are still some studies on this matter. The novelty of this work is mainly represented by the in vitro model that it is very important from a translational point of view, since it offers a useful tool for the study of new antithrombotic drugs.

However, I have some major concerns that need to be addressed before publication:

  • Figure 1B: the expected MW for the employed antibody is ~50 kDa. How do the authors explain the higher mw in the A2058 and, even more, in the MEG-01 cells?
  • Paragraph 2.1: authors stated that there is “no effective antibody against PAR-4”. From a rapid research, there is a series of companies (i.e. Abcam, Alomone, Santa Cruz) that sell anti-PAR-4 antibodies. How many antibodies did the authors try? I think that this is mandatory, since all the study is focused on PAR-4.
  • as regards the calcium levels, the authors described two different calcium dynamic upon treatment with PAR1-AP and PAR4-AP, respectively. Moreover, they correctly used thrombin to study calcium variations and they concluded that the thrombin activity is calcium-independent. Thus, which is the role of calcium, if any, in the PAR-1 and PAR-4 activated pathways? Why the intracellular increase is different between PAR-1 and PAR-4? It would be interesting to see if the morphological changes in MEG-01 occur even in the presence of calcium chelators (i.e. BAPTA) or of inhibitors of the calcium release (i.e. TMB-8 or 2-APB).

Please, revise also the text, since there are some typos.

Author Response

We greatly appreciate the editor’s and reviewers’ efforts to carefully review our manuscript and the valuable comments and suggestions offered for the improvement of the manuscript (ijms-1519617). We have made each of the suggested revisions. The points of criticism raised by the reviewers were addressed by a point-by-point response. Changes in the manuscript text are highlighted in red color font.

Reviewer #1:

Comments to the Author

The paper entitled “PAR4-mediated PI3K/Akt and RhoA/ROCK signaling pathways are essential for thrombin-induced morphological changes in MEG-01 cells” describes the role of the PAR-4 receptor in the platelet activation mechanism in a new in vitro model. The work is well-structured and properly conducted, even if in literature there are still some studies on this matter. The novelty of this work is mainly represented by the in vitro model that it is very important from a translational point of view, since it offers a useful tool for the study of new antithrombotic drugs. However, I have some major concerns that need to be addressed before publication.

  1. Figure 1B: the expected MW for the employed antibody is ~50 kDa. How do the authors explain the higher mw in the A2058 and, even more, in the MEG-01 cells?

Response: Thank you for the valuable comments. As mentioned by the reviewer, the expected MW of PAR1 is ~50 kDa, but since PAR1 is glycosylated at multiple sites, it is reported to be detected between 50 and 100 kDa depending on cell types [1,2]. Indeed, we believe that the higher band sizes of PAR1 in Figure 1B are due to the difference in glycosylation.

  1. Paragraph 2.1: authors stated that there is “no effective antibody against PAR-4”. From a rapid research, there is a series of companies (i.e. Abcam, Alomone, Santa Cruz) that sell anti-PAR-4 antibodies. How many antibodies did the authors try? I think that this is mandatory, since all the study is focused on PAR-4.

Response: Sorry for the confusion. We have tested with PAR4 antibodies from Santa Cruz, the most commonly used in Western blot against PAR4, and Abcam, but failed to detect PAR4 specific band. Although we did not show the Western blot for PAR-4 in this study, RT-PCR (Figure 1A) and calcium signaling by PAR4-AP and PAR4-specific inhibitor (Figure 2B) clearly showed the functional expression of PAR-4 in MEG-01 cells. In addition, the phrase "no effective antibody against PAR-4" was deleted form the revised manuscript to make it clear.

  1. As regards the calcium levels, the authors described two different calcium dynamic upon treatment with PAR1-AP and PAR4-AP, respectively. Moreover, they correctly used thrombin to study calcium variations and they concluded that the thrombin activity is calcium-independent. Thus, which is the role of calcium, if any, in the PAR-1 and PAR-4 activated pathways? Why the intracellular increase is different between PAR-1 and PAR-4? It would be interesting to see if the morphological changes in MEG-01 occur even in the presence of calcium chelators (i.e. BAPTA) or of inhibitors of the calcium release (i.e. TMB-8 or 2-APB).

Response: Thank you for the valuable comments. As reviewer’s comments, we observed the effect of inhibition of calcium release by 2-APB on PAR1- and PAR4-mediated morphological changes in MEG-01. Surprisingly, we found that PAR1-induced morphological change was almost completely suppressed by 2-APB, whereas PAR4-mediated morphological change was not affected or slightly enhanced by inhibition of IP3 receptor with 2-APB (Figure 6D, E, G). These results revealed that PAR1 and PAR4 regulate morphological changes through different signaling pathways in MEG-01 cells. Meanwhile, ionomycin triggered a sustained increase in intracellular calcium level (Figure 6C) but did not induce morphological changes (Figure 6F). These results indicated that morphological changes in MEG-01 cells are primarily regulated through the downstream signaling pathways of PAR1 and PAR4, rather than through the calcium signaling itself. These results described in the revised manuscript.

  1. Please, revise also the text, since there are some typos.

Response: Thank you for the comments. The manuscript has been carefully revised.

References

  1. AG, S.; TH, S.; B, C.; S, B.; IC, C.; T, K.; N, V.; J, T. N-linked glycosylation of protease-activated receptor-1 at extracellular loop 2 regulates G-protein signaling bias. Proceedings of the National Academy of Sciences of the United States of America 2015, 112, doi:10.1073/pnas.1508838112.
  2. AG, S.; J, T. N-linked glycosylation of protease-activated receptor-1 second extracellular loop: a critical determinant for ligand-induced receptor activation and internalization. The Journal of biological chemistry 2010, 285, doi:10.1074/jbc.M110.111088.

Reviewer 2 Report

In this study by Heo et al., was observed differential signaling pathways for thrombin-induced PAR1 and PAR4 activation in human megakaryoblastic leukemia cell line, MEG-01. It is an interesting article. Minor comments:

In the abstract: "...which were highly similar to thrombin- induced platelet signaling and morphological changes". Studies of PAR1 and PAR4 activation were performed on platelets?.

What physiological implications would it have thrombin-induced morphological change in MEG-01 cells?. 

Author Response

We greatly appreciate the editor’s and reviewers’ efforts to carefully review our manuscript and the valuable comments and suggestions offered for the improvement of the manuscript (ijms-1519617). We have made each of the suggested revisions. The points of criticism raised by the reviewers were addressed by a point-by-point response. Changes in the manuscript text are highlighted in red color font.

Reviewer #2:

Comments to the Author

In this study by Heo et al., was observed differential signaling pathways for thrombin-induced PAR1 and PAR4 activation in human megakaryoblastic leukemia cell line, MEG-01. It is an interesting article.

Minor comments:

  1. In the abstract: "...which were highly similar to thrombin- induced platelet signaling and morphological changes". Studies of PAR1 and PAR4 activation were performed on platelets?

Response: Sorry for the confusion. In this study, we used only MEG-01 cells to investigate the molecular mechanisms of thrombin induced morphological changes via PAR-1 and PAR-4. In abstract, the studies on PAR1 and PAR4 activation in human platelets have been reported in previous studies. Previous studies have shown that PAR1 and PAR4 expressed in human platelets exhibit different calcium dynamic [1]. In addition, it is very well known that thrombin induces shape changes in platelets mediated by both calcium-dependent signaling pathways and calcium-independent signaling pathways mainly involving RhoA activation [2,3]. In abstract, the sentence has been corrected for clear understanding.

  1. What physiological implications would it have thrombin-induced morphological change in MEG-01 cells?

Response: Thank you for the valuable comments. MEG-01 cells can differentiate into megakaryocytes (MKs) and produce platelet-like particles, which makes them an important tool for related research. The lack of research on the physiological role of thrombin receptors in MKs limits a clear understanding of the physiological role of PAR1 and PAR4 in MKs. However, during differentiation into MKs, MEG-01 cells show dynamic morphological changes with remodeling of cytoskeletal proteins such as actin, α-tubulin and β1-tubulin, which are prerequisites for proplatelet formation. Notably, thrombin treatment induces dramatic morphological changes in human MKs that are reminiscent of those found in platelets, such as shape changes and organelle centralization. In addition, it is well known that cytoskeletal remodeling of MKs and platelets is primarily regulated through the RhoA/ROCK signaling pathway. Interestingly, these results are consistent with our findings that PAR1-AP and PAR4-AP significantly induce morphological changes in MEG-01 cells (Figure 5), and thrombin-induced morphological changes in MEG-01 cells are primarily regulated by PAR4-mediated PI3K/Akt and RhoA/ROCK signaling pathways (Figure 7). Therefore, these results strongly suggest that PAR1 and PAR4 mediated signaling can play different roles not only in morphological changes but also in other related physiological events induced by thrombin in MKs. The above is described in the discussion section of the revised manuscript.

References

  1. MJ, S.; EJ, W.; TR, F.; SR, C. Protease-activated receptors 1 and 4 are shut off with distinct kinetics after activation by thrombin. The Journal of biological chemistry 2000, 275, doi:10.1074/jbc.M004589200.
  2. BZ, P.; JL, D.; SP, K. Platelet shape change is mediated by both calcium-dependent and -independent signaling pathways. Role of p160 Rho-associated coiled-coil-containing protein kinase in platelet shape change. The Journal of biological chemistry 1999, 274, doi:10.1074/jbc.274.40.28293.
  3. SL, B.; I, F.; M, G.; GF, N. Thrombin-induced activation of RhoA in platelet shape change. Biochemical and biophysical research communications 2001, 287, doi:10.1006/bbrc.2001.5547.

Round 2

Reviewer 1 Report

I thank the authors because they fully addressed all my requests. I think that the paper ameliorated and now it is suitable for publication.